# Study on Adding Ammonium Hydrogen Fluoride to Improve Manganese Leaching Efficiency of Ammonia Leaching Low-Grade Rhodochrosite

**Peng Yang, Xiaoping Liang \*, Chengbo Wu, Tengfei Cui and Yu Wang**

College of Materials Science and Engineering, Chongqing University, Chongqing 400030, China; yangpeng02101@163.com (P.Y.); wuchengbo@cqu.edu.cn (C.W.); cuitengfei2021@163.com (T.C.); wangyu@cqu.edu.cn (Y.W.)
\* Correspondence: xpliang@cqu.edu.cn; Tel.: +86-023-65127306

**Abstract:** The ammonia leaching method for treating low-grade rhodochrosite has the advantages of a good impurity removal effect and low environmental pollution. In this paper, aiming at the low leaching efficiency of low-grade rhodochrosite treated by the ammonia leaching method, studies on enhancing the leaching efficiency of manganese by using ammonium hydrogen fluoride as an additive are carried out. The effects of different ammonia concentrations, leaching temperatures, leaching times, liquid-solid ratios, stirring rates, and the addition of ammonium hydrogen fluoride on the leaching efficiency of manganese with and without ammonium hydrogen fluoride as an additive were comparatively studied, and the parameters of ammonia concentration, ammonia leaching temperature, and ammonium hydrogen fluoride dosage were optimized in the experimental study. The results indicated that ammonium hydrogen fluoride as an additive in the treatment of low-grade rhodochrosite by the ammonia leaching method could effectively increase the leaching efficiency of manganese, and the optimal process parameters were obtained. Meanwhile, the addition of ammonium hydrogen fluoride didn't affect the quality of the steamed ammonia product.

**Keywords:** rhodochrosite; ammonia leaching; ammonium hydrogen fluoride; leaching efficiency

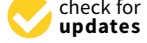



## 1. Introduction

Manganese (Mn) is one of the 12 most abundant elements (it comprises approximately 0.1% of the earth's crust), and can be associated in different ways, finding oxides, sulfides, carbonates, and silicates with greater abundance. Besides, some recent studies mention that the largest manganese reserves in the world (high-grade deposits) are found on the seabed [1]. At present, 95% of the manganese produced annually is being consumed by the steel industry, and the remaining 5% is being used by other industries, such as the chemical, paint, fertilizer, and battery industries [2]. Manganese levels in the world range from 20% to 60%, but a significant proportion belongs to the low- and medium-grade (Mn: 20–35%) category [3]. With the depletion of high-grade manganese resources, low-grade manganese is becoming more important.

Types of low-grade manganese include rhodochrosite, pyrolusite, alabandite, and so forth. When manganese is present in its bivalent soluble form, Mn, manganese salts are generally obtained directly by acid leaching. When manganese in ore is present in the form of manganese dioxide, since $MnO_2$ is stable in direct acid or alkali dissolution, the possible reduction of the insoluble tetravalent manganese to soluble bivalent manganese is necessary to recover manganese from its ores. In this regard, two processes are followed viz. reduction roasting followed by leaching and reductive leaching [4]. Zhang et al. [5] used chemical pure sulfur as a reducing agent to treat low-grade manganese oxide ore through reduction roasting and sulfuric acid leaching. The leaching efficiency was 95.6% for Mn, with a 550 °C roasting temperature, 10 min roasting time, 0.50 of S/Mn, 1.0 mol/L

sulfuric acid concentration, 25 °C leaching temperature, 200 r/min stirring rate, 5 min of leaching time, and a 5:1 liquid-to-solid ratio. Alaoui et al. [6] used potassium oxalate as a reducing agent in sulfuric acid medium to reduced and leached manganese in pyrolusite. The results show that when there is a sulfuric acid concentration of 0.25 to 1 M, oxalic acid concentration of 0.06, 0.09, 0.12, and 0.25 M, leaching temperature of 25 to 40 °C, and manganese ore particle size of −630, −280, −125, and −63 μm, the leaching efficiency of manganese increases with an increase of $H_2SO_4$ concentration, potassium oxalate concentration, leaching temperature, and decrease of particle size. Oxalic acid concentration has a great influence on Mn extraction, followed by temperature and sulfuric acid concentration. Prasetyo [7] studied the leaching of manganese from low-grade manganese ore in sulfuric acid mediums with tannic acid as a reducing agent. The results show that the optimum conditions for Mn leaching are tannic acid 40 g/L, sulfuric acid 0.5 M, liquid-solid ratio 5 mL/g, room temperature stirring 6 h, and the leaching efficiency of Mn increases with the increase of tannic acid and sulfuric acid concentration. Lu et al. [8] studied the process of manganese extraction from low-grade manganese oxide ore in sulfuric acid medium with formic acid as a reductant. The process with 15% $H_2SO_4$, 0.4 mL/g formic acid, a liquid-to-solid ratio of 6, and reaction at 90 °C for 2 h can result in a manganese leaching efficiency of up to 90.05%. It is known that rhodochrosite ($MnCO_3$) exhibits diamagnetic properties and has a magnetic susceptibility that is comparable with that of quartz or illite at room temperature. In addition, rhodochrosite can be converted to MnO by roasting. Thus, the treatment methods of low-grade rhodochrosite mainly include magnetic separation and acid leaching [9,10]. Although higher manganese leaching efficiency can be obtained by the above methods, there are some problems, such as high acid consumption, serious environmental pollution and equipment corrosion, and high impurity content. Compared with acid leaching for treating low-grade manganese ore, ammonia leaching makes manganese in manganese ore dissolve into manganese ammonia complexes during the ammonia leaching process, while impurity elements cannot dissolve in ammonia solution, which has good selectivity. Therefore, it is suitable for low-grade minerals with high impurity content, and the ammonia solution can be recycled without waste liquid.

The ammonia leaching method has been widely used in nickel, cobalt, zinc, copper minerals, or slag [11–13]. Chen et al. [14] used the reductive roasting-ammonia leaching method to extract nickel and cobalt from low-grade nickel-containing laterite ore. The leaching efficiency of nickel and cobalt reached 86.25% and 60.84%, respectively. At the same time, the leaching agent could also be recovered at normal temperature and pressure. Li et al. [15] treated low-grade zinc oxide minerals by irradiation roasting and ammonia leaching processes, and the leaching efficiency of zinc was able to reach 88.3%. Bidari [16] used the ammonia leaching method with ammonia and ammonium carbonate as leachables to extract 78% copper from copper smelter residue. It is thus concluded that ammonia leaching is well-suited for the treatment of low-grade minerals. At the same time, ammonia leaching has been regarded as one of the possible methods for manganese extraction by researchers because of its advantages, such as recycling of leaching agents, good selectivity to manganese, and less waste liquid production [17].

There are also some reports about leaching of manganese by the ammonia leaching method, such as Chen's [18] study of manganese extraction from permanganite by the ammonia leaching method, which reached up to 59.9–64.8%. Chen et al. [19] applied the ammonia leaching method to extract manganese from low-grade manganese, and the results showed that the ammonia to ammonium carbonate concentration ratio, leaching temperature, liquid-solid ratio and leaching time had significant effects on the leaching of manganese. The U.S. mineral administration [20] studied the ammonia leaching method for recovery of manganese from flat furnace residue, showing that manganese is dissolved in the form of its amine complexes. Mcintosh [21] studied the recovery of manganese from several BOF slags by the ammonia leaching method, showing that approximately 80% of manganese can be leached. From the current research situation, the research on ammonia leaching treatment of low-grade manganese ore is not deep enough. Our group [22]

employed the roasting ammonia leaching method to leach manganese from low-grade rhodochrosite, and the leaching efficiency of manganese could reach 85.6% under optimal conditions. Through the above studies, it was found that the ammonia leaching method has the deficiency of low manganese leaching efficiency. In order to solve this problem, this study aims to select an appropriate additive on the basis of previous studies to improve the leaching efficiency of manganese in the ammonia leaching process.

The addition of additives in the leaching process is an effective way to enhance the leaching efficiency of minerals, which has been confirmed by many studies. For example, Sun [23] used $Al_2(SO_4)_3$ and $Al_2O_3$ as additives in the leaching of coal system goethite, and Zhou [24] used lauryl alcohol as an additive in leaching potassium from phosphorus potassium-associated minerals, and found that the addition of additives enhanced the leaching of minerals equally well. Fluoride as an effective co-immersion agent can effectively improve the leaching efficiency of minerals [25]. Zhang et al. [26] used $NH_4F$ as an additive for leaching vanadium from sulfuric acid, and it was concluded that fluoride ions can promote the dissolution of poorly soluble aluminosilicate minerals, such as mica. Govindaiah [27] found that polytetrafluoroethylene is a promising additive for warm pressurised oxidative leaching in copper high sulfide, and the leaching efficiency and final extraction yield of copper can be significantly improved in the presence of polytetrafluoroethylene beads. Obviously, the addition of suitable additives in different mineral leaching processes can enhance the leaching efficiency of minerals. However, there is no report about adding additives in the leaching process of manganese by ammonia leaching.

In order to improve the manganese leaching efficiency of low-grade rhodochrosite treated by ammonia leaching, in this paper, ammonium hydrogen fluoride was selected as an additive during ammonia leaching treatment, and the effects of ammonium hydrogen fluoride with or without an additive on the leaching efficiency of manganese were comparatively studied using a single-factor method. The ammonia leaching process with the addition of ammonium hydrogen fluoride was also optimized by response surface methodology.

## 2. Material and Experimental Procedure

### 2.1. Material and Roasting Treatment

The experimental raw material in this study was taken from low-grade rhodochrosite in a region of Sichuan Province, China, which contained a 17.83% manganese element and a phosphomanganese ratio greater than 0.005. In addition, it also contains 12.6% calcium, 4.25% magnesium, 3.05% silicon, and other impurities. The main phases are manganese carbonate ($MnCO_3$), dolomite ($CaMg(CO_3)$), silica ($SiO_2$), and ferric carbonate ($FeCO_3$) [28].

In addition, the chemical reagents used in this study included ammonium carbamate with a purity of 99.0%, ammonium hydroxide with a concentration of 25%, and ammonium hydrogen fluoride with a purity of 99.0%.

Most manganese in low-grade rhodochrosite exists in the form of manganese carbonate, which cannot be dissolved directly in ammonia solution. Therefore, before ammonia leaching, the insoluble manganese carbonate needs to be converted to easily soluble MnO by roasting. The specific steps are as follows: 50.0 g rhodochrosite (with a particle size of 74 μm) is put into a drying box for two hours, and then placed in a high-temperature vertical tubular furnace. After roasting at 750 °C for 1 h, it is cooled to room temperature, and the whole process is carried out in a nitrogen atmosphere. The rhodochrosite after roasting was broken to 74 μm to obtain the roasted sand used in the ammonia leaching experiments. The main chemical compositions of the calcine were analyzed using ICP-OES (Thermo Fisher Scientific, Icap 6300 Duo, MA, USA), and the results are shown in Table 1. The phases of calcine were analyzed by X-ray (Panalytical, Empyrean, Almelo, The Netherlands) diffraction, and the results are shown in Figure 1.

**Table 1.** The analysis of the content of main elements in calcined manganese ore.

| Element | Mn | Ca | Si | Al | Mg | Fe | P | S |
|---|---|---|---|---|---|---|---|---|
| Content (w/%) | 23.19 | 9.41 | 4.15 | 1.30 | 2.73 | 1.24 | 0.51 | 0.51 |

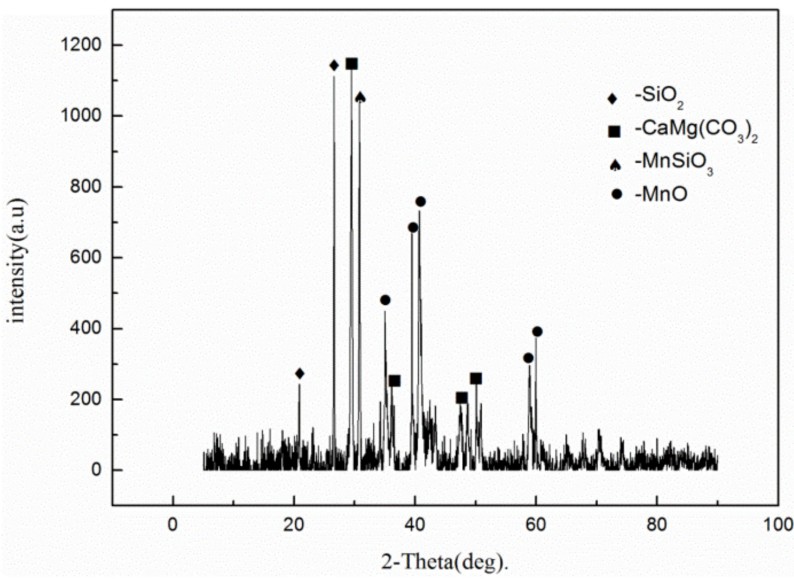

**Figure 1.** The XRD map of calcined manganese ore roasting sample.

*2.2. Experimental Method*

The specific experimental process of the low-grade rhodochrosite treatment process is shown in Figure 2: mainly including roasting, leaching, and evaporation. This paper focuses on the ammonia leaching process, and the influence of ammonium hydrogen fluoride as an additive on the leaching efficiency of manganese was mainly discussed by the single-factor variable method. When there is no ammonium hydrogen fluoride and the addition amount of ammonium hydrogen fluoride is 8%, the ammonia concentration varied from 6 mol/L to 14 mol/L, leaching temperature ranged from 25 °C to 35 °C, leaching time ranged from 0.5 h to 2.5 h, liquid-solid ratio varied from 3:1 to 6:1, and stirring rate varied from 200 r/min to 400 r/min, where five experiments were performed for each factor within the range of variation. The effect of the addition of ammonium hydrogen fluoride in the range of 2~10% on manganese leaching efficiency was further explored under the above experimental parameters. On this basis, the ammonia concentration, ammonia leaching temperature, and ammonium hydrogen fluoride addition amount were selected as three influencing factors of the experiment, and the response surface method was adopted to optimize the ammonia leaching process. See Section 3.3 for a detailed optimization experimental scheme.

The ammonia leaching process for manganese ore is not a simple use of ammonia water or ammonium salt solution as a leaching agent, but the use of an ammonia–ammonium salt mixed solution as a leaching agent. Because in pure ammonia solution, the manganese leaching efficiency is very low, by adding ammonium carbamate and ammonia to form a buffer solution to adjust the pH value of the solution, the pH value of the solution was suitable for the formation of a manganese–ammonia complex in this paper. The leaching reaction is shown as Equation (1).

$$MnO + xNH_3 + NH_2COO^- \rightarrow [MnO(NH_3)xNH_2COO]^- \tag{1}$$

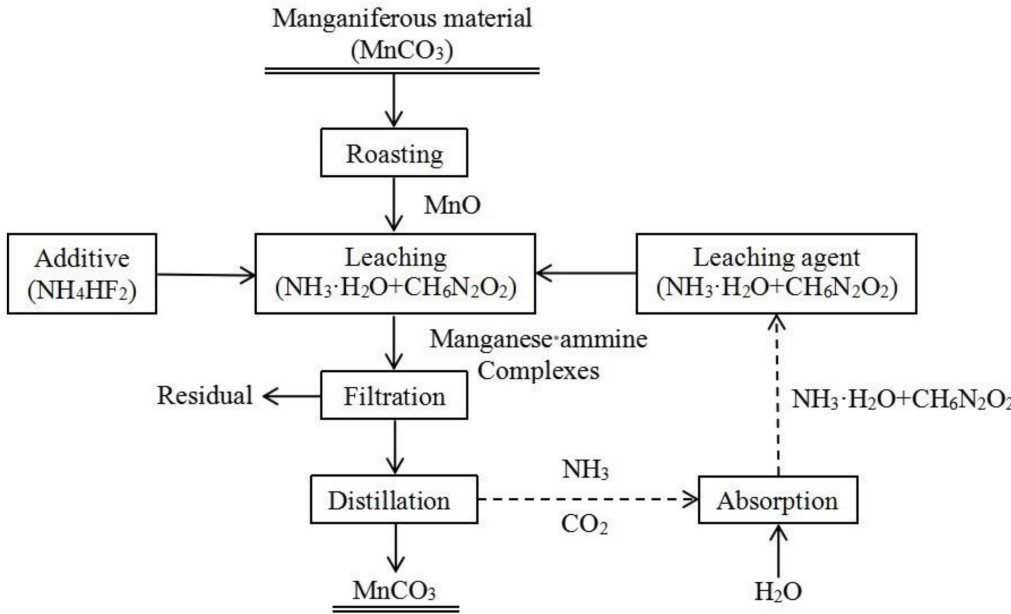

**Figure 2.** The technological process.

The specific steps of the ammonia immersion test are as follows: (1) A roasted low-grade rhodochrosite calcine sample for reserve, weighing 10 g; (2) the desired concentration of leaching agent was prepared using a 25% mass fraction of ammonia and analytical pure ammonium carbamate and added to a three-mouth flask (when the required ammonia concentration was greater than 13 mol/L, 25% of the ammonia solution was continuously injected with ammonia to achieve the desired ammonia concentration); (3) after the water bath temperature was set and the leaching agent reached the specified temperature, the low-grade rhodochrosite calcine sample was added; (4) the electric stirring paddle was opened and the sample of low-grade rhodochrosite calcine was dissolved at a certain stirring rate; (5) after leaching for a certain period of time, the solution was taken out and the solid and liquid were separated by vacuum extraction and filtration; (6) the volume of filtrate was measured and the content of manganese in filtrate was determined by ICP, and the phase of leaching residue was determined by XRD. The leaching efficiency of Mn was calculated by Equation (2):

$$\eta = \frac{m_L}{m_C} \times 100\% \tag{2}$$

$\eta$—Leaching efficiency of manganese, %; $m_L$—Mass of manganese in the leaching solution, g; $m_C$—Mass of manganese in the rhodochrosite calcine, g.

### 2.3. Preparation of Manganese Carbonate from Ammonia Leaching Solution

Because the manganese ammonia complexes bound to carbamate are very unstable, manganese is precipitated as manganese carbonate by steaming ammonia. The main reaction during evaporation is shown as Equation (3).

$$[MnO(NH_3)xNH_2COO]^- \rightarrow MnCO_3 + xNH_3 + CO_2 + OH^- \tag{3}$$

In order to investigate whether ammonium hydrogen fluoride as an additive in the ammonia leaching process has an impact on the final manganese product, the filtrate obtained from ammonia leaching was treated by ammonia evaporation, and the components of the manganese product were analyzed. The ammonia steaming temperature is 93 °C, and the ammonia steaming time is 60 min. The recovery of manganese during ammonia evaporation can be calculated by Equation (4):

$$\rho = \frac{m_P}{m_C} \times 100\% \tag{4}$$

ρ—Recovery of manganese, %; $m_P$—Mass of manganese in the product, g; $m_C$—Mass of manganese in rhodochrosite calcine, g.

## 3. Results and Discussion

Through the single-factor experimental results, the effects of ammonia concentration, leaching temperature, leaching time, liquid-solid ratio, stirring rate, and the amount of ammonium hydrogen fluoride on manganese leaching efficiency were analyzed; the ammonia leaching process was optimized by the response surface method; and the leach solution under the optimal ammonia leaching process was distilled with ammonia to determine whether the addition of ammonium hydrogen fluoride affects the quality of the final manganese product. The results are discussed below.

### 3.1. Effect of Additives on Manganese Leaching Efficiency

The effects of ammonia concentration, leaching temperature, leaching time, liquid-solid ratio and stirring rate on manganese leaching efficiency with or without ammonium hydrogen fluoride are shown in Figure 3. It can be seen from the figure that with or without the addition of ammonium hydrogen fluoride, the leaching efficiency of manganese always increases with the increase of ammonia concentration. Because the increase of ammonia concentration can promote the transformation of $Mn^{2+}$ into manganese ammonia complexes, the leaching efficiency of manganese increases sharply with the increase of leaching temperature, leaching time, and liquid-solid ratio, and then tends to be stable under certain conditions. Because the leaching reaction is exothermic, increasing the temperature is not conducive to leaching reactions; the leaching reaction tended to be balanced at 1.5 h, and the prolonged leaching time had little effect on the increase of leaching efficiency; the liquid-solid ratio was larger, and the more free ammonia and ammonium ions were in the solution, which accelerated the leaching reaction. When the liquid-solid concentration reaches 6:1, most of the manganese has been leached, and the leaching efficiency tends to be stable. The leaching efficiency of manganese firstly increased and then decreased with the increase in stirring rate. Due to the appropriate increase in the stirring rate, the external diffusion process of the reaction system was able to be accelerated, thus promoting the leaching reaction. However, when the stirring rate continues to increase, the volatilization of $NH_3$ will intensify, leading to the decrease of ammonia concentration and the decrease of Mn leaching efficiency. However, under the same conditions, the leaching efficiency of manganese with the addition of ammonium hydrogen fluoride is much higher than that without the addition of ammonium hydrogen fluoride. In conclusion, the leaching efficiency of manganese with ammonium hydrogen fluoride as an additive in the ammonia leaching process was greatly improved compared with that without ammonium hydrogen fluoride under any process parameters, indicating that ammonium hydrogen fluoride as an additive in the ammonia leaching of low-grade rhodochrosite calcine can promote the leaching of manganese. The single-factor experiment determined that when the addition of ammonium hydrogen fluoride was 8%, the optimum process parameters were a leaching temperature of 30 °C, ammonia concentration of 14 mol/L, liquid-solid ratio of 6:1, stirring rate of 400 r/min, and leaching time of 1 h.

### 3.2. Effect of Ammonium Hydrogen Fluoride Addition on Mn Leaching Efficiency

Under the optimum process parameters of leaching temperature of 30 °C, the ammonia concentration of 14 mol/L, liquid-solid ratio of 6:1, stirring rate of 400 r/min, and leaching time of 1 h, the effect of the ammonium hydrogen fluoride additive amount on manganese leaching efficiency was investigated, and the results are shown in Figure 4. It can be seen from the figure that, when the addition amount of ammonium hydrogen fluoride is increased from 2% to 6%, the leaching efficiency of manganese is increased from 84.7% to 91.6%, and then remains basically unchanged with the increase of the addition amount. It can be seen that with the increase of the addition of ammonium hydrogen fluoride,

manganese silicate in rhodochrosite calcine is continuously decomposed, thus increasing the leaching efficiency of manganese.

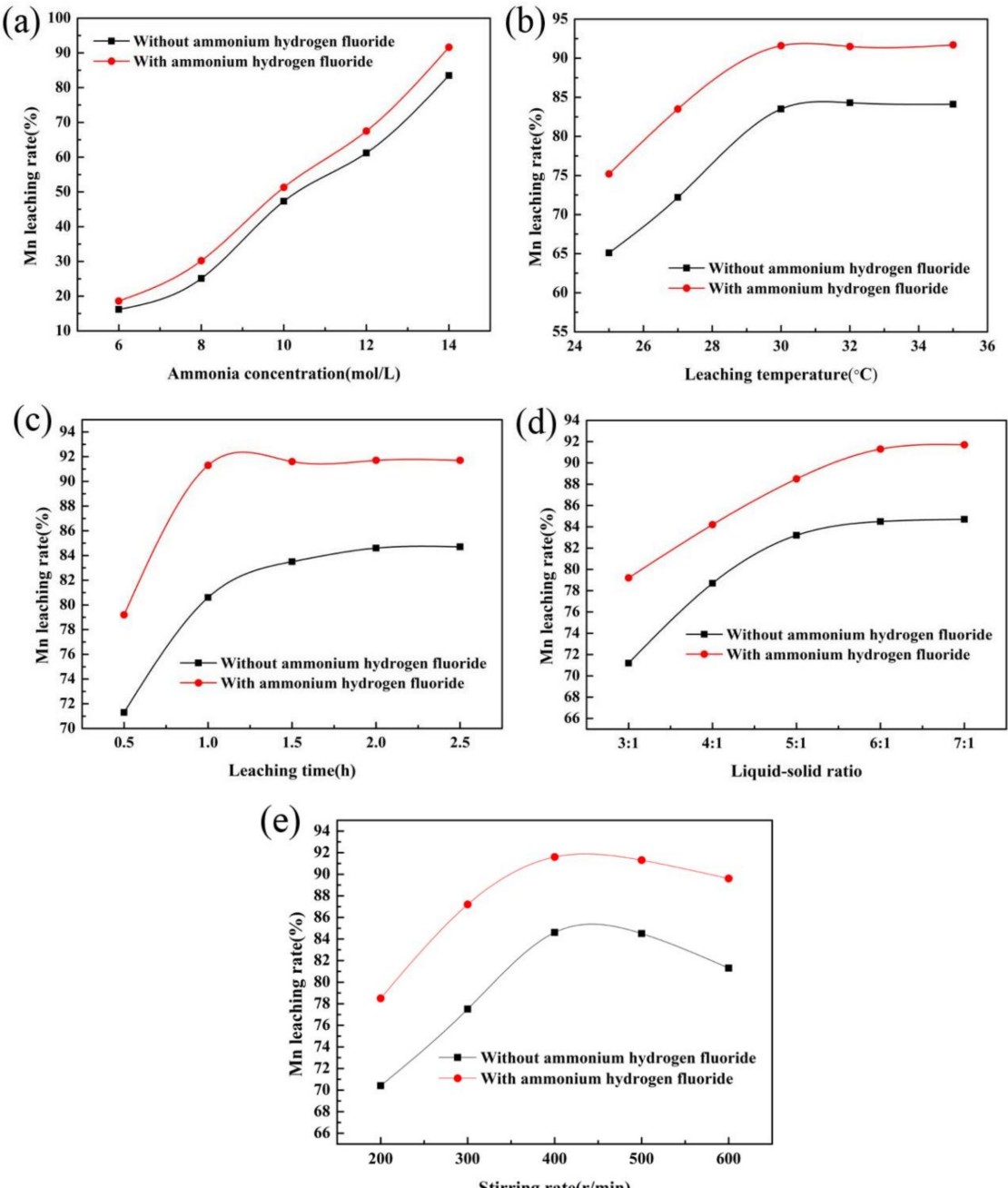

**Figure 3.** Effects of different factors on manganese leaching efficiency with or without annonium hydrogen fluoride. (**a**) Leaching temperature of 30 °C, liquid–solid ratio of 6:1, stirring rate of 500 r/min, leaching 1.5 h, ammonium hydrogen fluoride dosage of 8% (and 0%); (**b**) ammonia concentration of 14 mol/L, liquid–solid ratio of 6:1, stirring rate of 500 r/min, leaching time of 1.5 h, amount of ammonium hydrogen fluoride of 8% (and 0%). (**c**) Leaching temperature of 30 °C, ammonia concentration of 14 mol/L, liquid–solid ratio of 6:1, stirring rate of 500 r/min, ammonium hydrogen fluoride dosage of 8% (and 0%); (**d**) leaching temperature of 30 °C, ammonia concentration of 14 mol/L, stirring rate of 500 r/min, leaching time of 1 h, amount of ammonium hydrogen fluoride of 8% (and 0%). (**e**) Leaching temperature of 30 °C, ammonia concentration of 14 mol/L, liquid–solid ratio of 6:1, leaching time of 1 h, amount of ammonium hydrogen fluoride of 8% (and 0%).

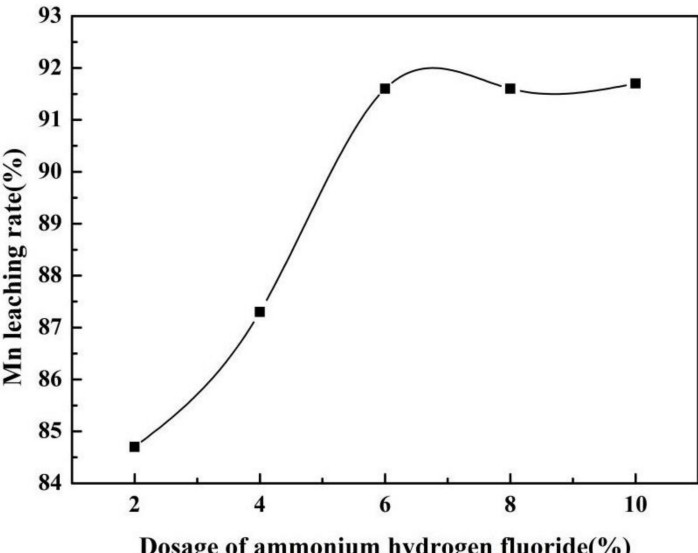

**Figure 4.** Leaching temperature of 30 °C ammonia concentration of 14 mol/L, liquid–solid ratio of 6:1, stirring rate of 400 r/min, and leaching time of 1 h, and the influence of the addition amount of ammonium hydrogen fluoride on the leaching efficiency of Mn.

In conclusion, the suitable leaching conditions obtained from ammonia leaching of rhodochrosite calcine with ammonium hydrogen fluoride are as follows: The addition of ammonium hydrogen fluoride is 6%, the ammonia leaching temperature is 30 °C, the ammonia leaching time is 1 h, the liquid-solid ratio is 6:1, the ammonia concentration is 14 mol/L, and the stirring rate is 400 r/min. Under these conditions, the leaching efficiency of Mn in rhodochrosite calcine can reach 91.6%, while under the same conditions, the leaching efficiency of manganese without ammonium hydrogen fluoride is only 84.6%. Therefore, the addition of ammonium hydrogen fluoride can improve the leaching efficiency of manganese, so as to strengthen the ammonia leaching process of rhodochrosite calcine.

We can see from the experimental results that the leaching efficiency of manganese is improved after adding ammonium hydrogen fluoride. As for the mechanism of ammonium hydrogen fluoride on improving manganese leaching efficiency in the ammonia leaching process of rhodochrosite calcine, we first calculated the phase composition balance of solution in the ammoniacal leaching process of ammonium hydrogen fluoride in the rhodochrosite calcine under the $[NH_3]_T = 10$ mol/L, $[Mn^{2+}]_T = 1$ mol/L, $[F^-]_T = 0.1$ mol/L by using HYDRA/MEDUSA software. It was found that the addition of fluorine will not combine with $Mn^{2+}$ to form a complex under the experimental conditions, so it is considered that the F element may affect the manganese leaching process in other ways. We tried to find an explanation from the phase of leach residue with or without the addition of ammonium hydrogen fluoride. By comparing the XRD patterns of leached slag with and without ammonium hydrogen fluoride in Figure 5, we found that $MnSiO_3$ was missing in the XRD patterns of leached slag with ammonium hydrogen fluoride added. Through a literature investigation, it was found that fluoride ions can promote the destruction of mica minerals when fluoride is added in the process of mineral leaching [29–31]. Therefore, it is speculated that ammonium hydrogen fluoride has an erosive effect on the manganese silicate in the calcine of rhodochrosite, which makes the manganese enter into the solution and improve the leaching efficiency of Mn in the calcine of rhodochrosite.

*3.3. Process Optimization of Ammonia Leaching of Rhodochrosite Calcine with the Addition of Ammonium Hydrogen Fluoride*

The central combined test design (CCD) was carried out with the manganese leaching efficiency as the response value [32]. The level of experimental factors is shown in Table 2.

Twenty experiments were performed according to Table 2, and the experimental results are shown in Table 3.

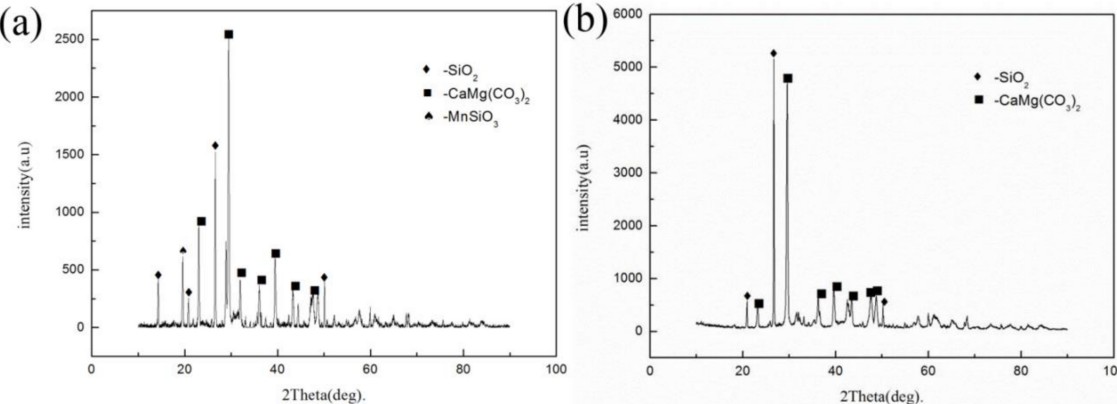

**Figure 5.** (**a**) Without ammonium fluoride in XRD diagram of slag leaching; (**b**) with ammonium fluoride in XRD diagram of slag leaching.

**Table 2.** The ranges and levels of CCD variables in ammoniacal leaching of roasted manganese ore by ammonium fluoride.

| Investigate Factors | Symbol | Level | | | | |
|---|---|---|---|---|---|---|
| | | $\alpha = -1.682$ | $-1$ | $0$ | $1$ | $\alpha = +1.682$ |
| Ammonia leaching temperature | °C | 26.64 | 28 | 30 | 32 | 33.36 |
| Ammonia concentration | mol/L | 6.64 | 8 | 10 | 12 | 13.36 |
| Ammonium hydrogen fluoride addition | w/% | 2.64 | 4 | 6 | 8 | 9.36 |

**Table 3.** The experimental scheme and results of CCD variables in ammoniacal leaching of the roasted manganese ore by ammonium fluoride.

| Number | Ammonia Concentration (mol/L) | Ammonia Leaching Temperature (°C) | Ammonium Hydrogen Fluoride Addition (w/%) | Leaching Efficiency (%) |
|---|---|---|---|---|
| 1 | 10.00 | 33.36 | 6.00 | 52.4 |
| 2 | 10.00 | 30.00 | 2.64 | 49.3 |
| 3 | 13.36 | 30.00 | 6.00 | 90.6 |
| 4 | 10.00 | 30.00 | 6.00 | 51.3 |
| 5 | 12.00 | 32.00 | 4.00 | 64.9 |
| 6 | 10.00 | 30.00 | 6.00 | 51.3 |
| 7 | 12.00 | 28.00 | 8.00 | 66.8 |
| 8 | 6.64 | 30.00 | 6.00 | 23.3 |
| 9 | 8.00 | 32.00 | 8.00 | 31.2 |
| 10 | 8.00 | 28.00 | 8.00 | 30.7 |
| 11 | 8.00 | 32.00 | 4.00 | 27.1 |
| 12 | 10.00 | 30.00 | 6.00 | 51.3 |
| 13 | 10.00 | 30.00 | 6.00 | 51.3 |
| 14 | 8.00 | 28.00 | 4.00 | 26.3 |
| 15 | 10.00 | 30.00 | 6.00 | 51.3 |
| 16 | 10.00 | 26.64 | 6.00 | 48.9 |
| 17 | 12.00 | 32.00 | 8.00 | 68.4 |
| 18 | 10.00 | 30.00 | 9.36 | 54.2 |
| 19 | 12.00 | 28.00 | 4.00 | 63.8 |
| 20 | 10.00 | 30.00 | 6.00 | 51.3 |

The data in Table 3 were fitted by multiple regression. Manganese leaching efficiency is the dependent variable (Y), and ammonia concentration (A, mol/L), ammonia leaching temperature (B, °C) and ammonium hydrogen fluoride dosage (C, %) are the independent

variables. Through the least square fitting, the quadratic multiple regression equation of the rhodochrosite calcine ammonia leaching process can be obtained as follows:

$$Y = 51.63 + 20.38 \times A + 0.48 \times B + 0.89 \times C - 0.037 \times A \times B + 0.31 \times A \times C - 0.062 \times B \times C + 3.093E - 003 \times A^2 - 1.48 \times B^2 - 1.68 \times C^2 \quad (5)$$

The accuracy of the model can be further tested by analysis of variance, the significance of all coefficients in the polynomial equation can be obtained, and the effectiveness of the model can be judged. The regression analysis of variance obtained in this experiment is shown in Table 4.

**Table 4.** Analysis of variance for response surface quadratic model.

| Sources of Variance | Sum of Squares | df | Mean Square | F Value | *p*-Value Prob > F | Significance |
|---|---|---|---|---|---|---|
| Model | 5754.23 | 9 | 639.36 | 43.07 | <0.0001 | significant |
| A | 5672.43 | 1 | 5672.43 | 382.13 | <0.0001 | |
| B | 3.16 | 1 | 3.16 | 3.21 | 0.0345 | |
| C | 10.91 | 1 | 10.91 | 10.74 | 0.0013 | |
| AB | 0.011 | 1 | 0.011 | 0.012 | 0.9786 | |
| AC | 0.78 | 1 | 0.78 | 0.053 | 0.8232 | |
| BC | 0.031 | 1 | 0.031 | 0.021 | 0.9643 | |
| $A^2$ | $1.379 \times 10^{-4}$ | 1 | $1.379 \times 10^{-4}$ | 1.025 | 0.0006 | |
| $B^2$ | 31.64 | 1 | 31.64 | 2.13 | 0.1750 | |
| $C^2$ | 40.49 | 1 | 40.49 | 2.73 | 0.1296 | |
| Residual | 148.44 | 10 | 14.84 | | | |
| Lack of Fit | 148.25 | 5 | 29.65 | 787.19 | <0.0001 | significant |
| Pure Error | 0.19 | 5 | 0.038 | | | |
| Cor Total | 5902.67 | 19 | | | | |

It can be seen from Table 4 that the F value of the model is 43.07, the model is significant, and only 0.01% probability will make the signal-to-noise ratio error. The value of "prob > F" less than 0.0500 indicates that the model item is significant. Under this model, A, B, C, and $A^2$ have significant effects on manganese leaching efficiency, while AB, BC, AC, $B^2$, and $C^2$ have non-significant effects on manganese leaching efficiency. The analysis of variance shows that the model fits well with the experimental data, and can accurately predict the ammonia leaching process of rhodochrosite calcine.

According to the regression model, the theoretically optimized process parameters of ammonia concentration, ammonia leaching temperature, and ammonium hydrogen fluoride dosage were calculated, as shown in Table 5. In order to verify the accuracy of the model, the experiment was carried out to optimize the ammonia leaching process parameters. The manganese leaching efficiency was 92.0%, and the deviation from the predicted value was only 1.20%, which shows that the response surface method is reliable to optimize the process parameters of ammonia leaching of rhodochrosite calcine added with ammonium hydrogen fluoride.

**Table 5.** Optimization process parameters of the regression model.

| Ammonia Concentration (mol/L) | Ammonia Leaching Temperature (°C) | Ammonium Hydrogen Fluoride Addition (w/%) | Leaching Efficiency Predicted Value (%) | Leaching Efficiency Experiment Value (%) |
|---|---|---|---|---|
| 14.5 | 31.6 | 7.3 | 93.2 | 92.0 |

### 3.4. Characterization of Ammonia Products

In order to explore whether ammonium hydrogen fluoride as an additive in the ammonia leaching process has an impact on the final manganese product, the leaching solution obtained under the best process of adding additives was treated by ammonia evaporation. The obtained product was a light brown powder, and the composition analysis is shown in Table 6. The grade of manganese reached 44.13% and the recovery of manganese was 97.3%. The ammonia products were dried in the oven for XRD analysis,

and the results are shown in Figure 6. The main component of the ammonia steaming product was manganese carbonate, and there were a few miscellaneous peaks, indicating that the main component of the product obtained through the ammonia steaming process is manganese carbonate, and the ammonium hydrogen fluoride as an additive in the ammonia leaching process has no effect on the final manganese product.

**Table 6.** Quality analysis results of manganese carbonate products.

| Main Components | Mn | Ca | Mg |
|---|---|---|---|
| Content/% | 44.13 | 1.31 | 1.80 |

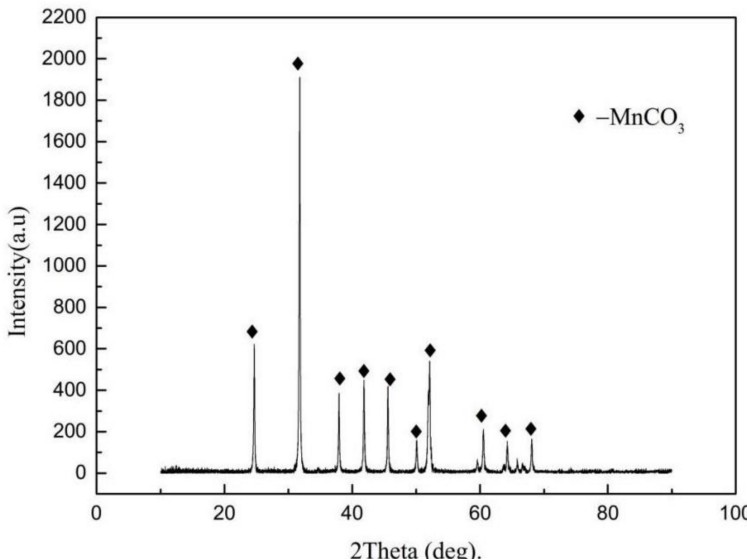

**Figure 6.** XRD of manganese carbonate in ammonia vapor product.

## 4. Conclusions

In this paper, the effects of ammonia concentration, leaching temperature, leaching time, liquid-solid ratio, stirring rate, and the amount of ammonium hydrogen fluoride on manganese leaching efficiency were discussed; in the range of experimental parameters, the ammonia concentration, ammonia leaching temperature, and the amount of ammonium hydrogen fluoride were optimized. The following conclusions are drawn:

1. As an additive in the ammonia leaching process, ammonium hydrogen fluoride can effectively improve the leaching efficiency of manganese, which is 7% higher than that without ammonium hydrogen fluoride.
2. The optimum technological conditions are as follows: ammonia concentration is 14.5 mol/L, ammonia leaching temperature is 31.6 °C, the dosage of ammonium hydrogen fluoride is 7.3%, ammonia leaching time is 60 min, liquid-solid ratio is 6:1, and stirring rate is 400 r/min.
3. The addition of ammonium hydrogen fluoride will not affect the quality of ammonia evaporation products, and the recovery of manganese is as high as 97.3%. The manganese grade of the manganese carbonate product is 44.13%.

It was found that adding ammonium hydrogen fluoride as an additive can effectively improve the leaching efficiency of manganese in the ammonia leaching process of low-grade rhodochroite. Therefore, it provides an alternative method for industrial treatment of low-grade rhodochroite. However, this method has the disadvantage of accumulation of waste liquid-containing fluorine. How to efficiently treat or recycle the waste liquid-containing fluorine is a challenge for the industrial application of this method, and is also a possible work to be considered in the future.

**Author Contributions:** P.Y. contributed in research and wrote the paper; X.L. contributed in the review and editing; Y.W., T.C. and C.W. contributed in resources. All authors have read and agreed to the published version of the manuscript.

**Funding:** This research received no external funding.

**Conflicts of Interest:** The authors declare they have no conflict of interest.

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
