# Peer review of "Study on Adding Ammonium Hydrogen Fluoride to Improve Manganese Leaching Efficiency of Ammonia Leaching Low-Grade Rhodochrosite"

_metals, doi:10.3390/met11091496_

Round 1
Reviewer 1 Report
The paper describes an interesting aspect of improvement on the overall leaching of manganese from a low-grade manganese ore. The followings are comments that the authors may like to respond to before the final recommendation for acceptance to be published in this Journal.
Lines 34 and 35: repeated sentences
Line 38: The authors seem to use the leaching rate and the percent recovery in the same manner, but this has to be distinguished. It is true that the leaching rate and the recovery are related but they are different terms used in extractive metallurgy. For example, rate is given in moles/l/s and recovery is given in %.
Line 38: When Alaoui is mentioned, the reference should be given. The same thing happens in the subsequent references. The authors have the habit of listing references prior to the names of the authors, which is confusing to the readers. For example, it is suggested by Alaoui et. al. (2006). They have shown in their experiments …
Lines 110 and 111: The sudden appearance of rhodochromite is puzzling. What is the relationship between this mineral and rhodochrocite? Is it a part of the host rock? Does it have any bearing on the leaching behavior? Are the authors suggesting that rhodochromite=rhodochrocite?
Fig 2: Filtration of the first part and Distillation of the second part are disconnected. Is there any reason for that? The process line for fluoride is not indicated. What do the authors suggest in taking care of the fluoride accumulation in the system? This could be a problematic issue for the process.
Eq 1 : Define mT and mg. This equation seems to represent the percent recovery and not the rate of reaction! The same comment is extended to Eq. 2. Please clarify!
The authors should pay attention to the use of singular and plural. It is quite confusing for the readers to follow. The paper should be edited by a native English-speaking person.
Fig. 5: The conclusion drawn based on this figure is not very convincing. The authors should make more study to make such a conclusion. It is only a weak speculation, and one should be more responsible to arrive at such a statement.
Overall Comment:
The authors have made a study to improve the overall recovery of manganese by adding ammonium hydrogen fluoride. This reviewer believes that the study would have been better, should the authors have clarified the chemistry involved on the role of ammonium hydrogen fluoride. Is it the fluoride ion responsible for the improvement? Does it matter if one uses other fluoride compounds to the system to arrive at the similar effect? The authors should have discussed the handling of fluoride after the leaching. The authors seem to imply that fluoride might have taken out manganese silicate that is responsible for the improvement on the leaching recovery. If it is the intention, the authors should have given stronger evidences to prove this is the case. This could have been the theme of the paper.
Reviewer 2 Report
Dear Authors
The present study called "Study on Adding Ammonium Hydrogen Fluoride to Improve Manganese Leaching Rate of Ammonia Leaching Low-Grade Rhodochrosite" presents interesting results. However, the introduction should be considerably improved: deepen the scientific discussion on the results, and also improve minor details. In addition, they should highlight the scientific novelty and importance of this work.
I think it may be an interesting manuscript for readers, after Major Revisions.
Page 1. Line 26-32. I think I should mention the abundance of manganese in the world (with figures). For example: “Manganese (Mn) is one of the twelve most abundant elements (it comprises approximately 0.1% of the earth’s crust” DOI: 10.1016/j.mineng.2020.106748
Also, some recent studies mention that the largest manganese reserves in the world (high grade deposits) are found on the seabed.
Page 1. Line 36. Correct the citation, the correct thing is Zang et al [3] used chemical pure…
Correct this, throughout the document. This error is repeated.
Page 1. Line 38-41. Here it is mentioned that an increase in the concentration of H2SO4 and reducing agent improves the extraction of manganese. But this statement is incomplete. Various studies mention that when working at high concentrations of reducing agent, the concentration of sulfuric acid becomes irrelevant (DOI: 10.1016/j.hydromet.2007.07.003; DOI: 10.3390/met9080903; DOI: 10.1016/j.mineng.2020.106748) .
Page 1. Line 41-44. It must indicate at what concentration of reducing agent was used in the mentioned study. Otherwise, the conclusion mentioned regarding acid would not be understood.
Page 2. Line 45-47. You must indicate figures so that it is understood how 90.08% of Mn was obtained. For example, acid concentration 2 mol / L, temperature ?, etc
Page 2 Line 64-65. "Ammonia leaching method has been regarded by researchers as a possible option for the extraction of manganese" Why is it cheaper as a reactive? Less polluting? Can it be reused? Does it present better extractions than other processes? You should mention this more precisely.
Page 3. Line 97-102. It should state the purpose of this work, and why it is scientifically novel.
Separate in the whole document, the figures of the units. For example, 50g…. The correct thing is 50 g.
Also, you should mention Figures throughout the document in capital letters. For example, Figure 1.
Figure 2 has errors, it should be reviewed and improved.
Page 5. Line 184-187. Can this conclusion be compared with the results of a previous study?
In general, the results are correct. But you must compare or ratify them based on previous studies. This would improve the quality of the discussion of your own results.
The conclusions are correct and precise. I think that after these, I could mention your next study on this subject, or the barriers to overcome to apply it industrially in the future.
Regards
Reviewer 3 Report
The manuscript deals with the investigation on the effect of Ammonium Hydrogen Fluoride to Improve Manganese Leaching Rate of Ammonia Leaching Low-Grade Rhodochrosite. The language of the manuscript needs major improvement. In the present form, the manuscripts do not provide much interest to the readers and need major revision before its resubmission to the journal. The authors are advised to incorporate the below recommendations:
- Kindly provide the proper reference for the statement “At present, the treatment methods of low-grade manganese mainly include acid leaching and ammonia leaching.
- Correct the spelling of manganese in lines 29, 30, 34, 35, and 69.
- The manuscript deals with the recovery of manganese from low-grade rhodochrosite whereas the introduction doesn’t report any investigation related to rhodochrosite. Authors are suggested to include a brief review on rhodochrosite processing. Please revise the introduction.
- Line 110 “low-grade rhodochromite”. Similarly in line 111.
- Why roasting has been performed before the ammonia leaching? Please explain. As the experiment has been carried with the roasted material, please replace the term “low-grade rhodochrosite” with “roasted low-grade rhodochrosite” in the results and discussion section.
- Redraw figure 2.
- Line 144 “ammonium carbamate”? please correct.
- Why ammonium carbonate has been added to the ammonia solution before leaching. Specify the reason.
- Authors are suggested to provide the mechanism of leaching.
- Please explain how and why the addition of ammonium hydrogen fluoride improved the recovery of manganese. Present the chemical reaction.
- Please provide the complete experimental condition in the caption for each plot (Figure 3, 4) for better understanding.
- Section 3.1 appears to be written superficially without proper and scientific explanation.
- Line 205 “rhododenite calcine”?
- Provide the complete chemical composition of leach solution obtained at optimized conditions.
- What is the significance of the 7% recovery difference?
Round 2
Reviewer 1 Report
The authors have responded to the comments made by the reviewer but there are a couple of points should be made. The authors should change the following before this paper can be accepted for publication. There are two formulae and two equations but they all should be changed to equations and therefore there should be Equations 1, 2, 3 and 4 instead of Formula 1 and 2 and Equations 1 and 2.
For the future publication, the authors should go back to some fundamental definitions used in the field of hydrometallurgy, the reaction rate is not described in terms of %! The authors should understand that the rate has an implication of time in it. For example, the rate is given in mol/l/sec. and never in %!!
Reviewer 2 Report
Dear Authors
Suggested corrections were made, and the manuscript has greatly improved.
The references [1] and [4] must be corrected, the full surnames and the names must be in initials (it is backwards).
Also, I recommend mentioning a possible future work at the end of the conclusions, and mentioning the problems and / or challenges for a possible industrial work.
Regards
Reviewer 3 Report
The authors have addressed satisfactorily the technical issues raised by the reviewers and read better than the original. The Manuscript may be accepted for publication.
Please correct the unit of particle size (line 142).
Please report the recovery to three significant figures (e.g. 97.3, instead of 97.31).
